# PoweredSGD: Powered Stochastic Gradient Descent Methods for Accelerated Non-Convex Optimization

## Abstract

We propose a novel technique for improving the stochastic gradient descent (SGD) method to train deep networks, which we term *PoweredSGD*. The proposed PoweredSGD method simply raises the stochastic gradient to a certain power $\gamma \in [0, 1]$ elementwise during iterations and introduces only one additional parameter, namely, the power exponent $\gamma$ (when $\gamma = 1$, PoweredSGD reduces to SGD). We further propose PoweredSGD with momentum, which we term *PoweredSGDM*, and provide convergence rate analysis on both PoweredSGD and PoweredSGDM methods. Experiments are conducted on popular deep learning models and benchmark datasets. Empirical results show that the proposed PoweredSGD and PoweredSGDM obtain faster initial training speed than adaptive gradient methods, comparable generalization ability with SGD, and improved robustness to hyper-parameter selection and vanishing gradients. PoweredSGD is essentially a gradient modifier via a nonlinear transformation. As such, it is orthogonal and complementary to other techniques for accelerating gradient-based optimization.

## 1 Introduction

Stochastic optimization as an essential part of deep learning has received much attention from both the research and industry communities. High-dimensional parameter spaces and stochastic objective functions make the training of deep neural network (DNN) extremely challenging. Stochastic gradient descent (SGD) (Robbins & Monro, 1951) is the first widely used method in this field. It iteratively updates the parameters of a model by moving them in the direction of the negative gradient of the objective evaluated on a mini-batch. Based on SGD, other stochastic optimization algorithms, e.g., SGD with Momentum (SGDM) (Qian, 1999), AdaGrad (Duchi et al., 2011), RMSProp (Tieleman & Hinton, 2012), Adam (Kingma & Ba, 2015) are proposed to train DNN more efficiently.

Despite the popularity of Adam, its generalization performance as an adaptive method has been demonstrated to be worse than the non-adaptive ones. Adaptive methods (like AdaGrad, RMSProp and Adam) often obtain faster convergence rates in the initial iterations of training process. Their performance, however, quickly plateaus on the testing data (Wilson et al., 2017). In Reddi et al. (2018), the authors provided a convex optimization example to demonstrate that the exponential moving average technique can cause non-convergence in the RMSProp and Adam, and they proposed a variant of Adam called AMSGrad, hoping to solve this problem. The authors provide a theoretical guarantee of convergence but only illustrate its better performance on training data. However, the generalization ability of AMSGrad on test data is found to be similar to that of Adam, and a considerable performance gap still exists between AMSGrad and SGD (Keskar & Socher, 2017; Chen et al., 2018). Indeed, the optimizer is chosen as SGD (or with Momentum) in several recent state-of-the-art works in natural language processing and computer vision (Luo et al., 2018; Wu & He, 2018), where in these instances SGD does perform better than adaptive methods. Despite the practical success of SGD, obtaining sharp convergence results in the non-convex setting for SGD to efficiently escape saddle points (i.e., convergence to second-order stationary points) remains a topic of active research (Jin et al., 2019; Fang et al., 2019).

**Related Works:** SGD, as the first efficient stochastic optimizer for training deep networks, iteratively updates the parameters of a model by moving them in the direction of the negative gradient of the

objective function evaluated on a mini-batch. SGDM brings a Momentum term from the physical perspective, which obtains faster convergence speed than SGD. The Momentum idea can be seen as a particular case of exponential moving average (EMA). Then the adaptive learning rate (ALR) technique is widely adopted but also disputed in deep learning, which is first introduced by AdaGrad. Contrast to the SGD, AdaGrad updates the parameters according to the square roots of the sum of squared coordinates in all the past gradients. AdaGrad can potentially lead to huge gains in terms of convergence (Duchi et al., 2011) when the gradients are sparse. However, it will also lead to rapid learning rate decay when the gradients are dense. RMSProp, which first appeared in an unpublished work (Tieleman & Hinton, 2012), was proposed to handle the aggressive, rapidly decreasing learning rate in AdaGrad. It computes the exponential moving average of the past squared gradients, instead of computing the sum of the squares of all the past gradients in AdaGrad. The idea of AdaGrad and RMSProp propelled another representative algorithm: Adam, which updates the weights according to the mean divided by the root mean square of recent gradients, and has achieved enormous success. Recently, research to link discrete gradient-based optimization to continuous dynamic system theory has received much attention (Yuan et al., 2016; Mazumdar & Ratliff, 2018). While the proposed optimizer excels at improving initial training, it is completely complementary to the use of learning rate schedules (Smith & Topin, 2019; Loshchilov & Hutter, 2016). We will explore how to combine learning rate schedules with the PoweredSGD optimizer in future work.

While other popular techniques focus on modifying the learning rates and/or adopting momentum terms in the iterations, we propose to modify the gradient terms via a nonlinear function called the Powerball function by the authors of Yuan et al. (2016). In Yuan et al. (2016), the authors presented the basic idea of applying the Powerball function in gradient descent methods. In this paper, we 1) systematically present the methods for stochastic optimization with and without momentum; 2) provide convergence proofs; 3) include experiments using popular deep learning models and benchmark datasets. Another related work was presented in Bernstein et al. (2018), where the authors presented a version of stochastic gradient descent which uses only the signs of gradients. This essentially corresponds to the special case of PoweredSGD (or PoweredSGDM) when the power exponential $\gamma$ is set to 0. We also point out that despite the name resemblance, the power PowerSign optimizer proposed in Bello et al. (2017) is a conditional scaling of the gradient, whereas the proposed PoweredSGD optimizer applies a component-wise trasformation to the gradient.

**Contributions:** Inspired by the Powerball method in Yuan et al. (2016), this paper uses Powerball-based stochastic optimizers for the training of deep networks. In particular, we make the following major contributions:

1. We propose the PoweredSGD, which is the first systematic application of the Powerball function technique in stochastic optimization. PoweredSGD simply applies the Powerball function (with only one additional parameter $\gamma$) on the stochastic gradient term in SGD. Hence, it is easy to implement and requires no extra memory. We also propose the PoweredSGDM as a variant of PoweredSGD with momentum to further improve its convergence and generalization abilities.

2. We have proved the convergence rates of the proposed PoweredSGD and PoweredSGDM. It has been shown that both the proposed PoweredSGD and PoweredSGDM attain the best known rates of convergence for SGD and SGDM on non-convex functions. In fact, to the knowledge of the authors, the bounds we proved for SGD and SGDM (as special cases of PoweredSGD and PoweredSGDM when $\gamma = 1$) provide the currently best convergence bounds for SGD and SGDM in the non-convex setting in terms of both the constants and rates of convergence (see, e.g. Yan et al. (2018)).

3. Experimental studies are conducted on multiple popular deep learning tasks and benchmark datasets. The results empirically demonstrate that our methods gain faster convergence rate especially in the early train process compared with the adaptive gradient methods. Meanwhile, the proposed methods show comparable generalization ability compared with SGD and SGDM.

**Outline:** The remainder of the paper is organized as below. Section 2 proposes the PoweredSGD and PoweredSGDM algorithms. Section 3 provides convergence results of the proposed algorithms for non-convex optimization. Section 4 gives the experiment results of the proposed algorithms on a variety of models and datasets to empirically demonstrate their superiority to other optimizers. Finally, conclusions are drawn in section 5.

**Notation:** Given a vector $a \in \mathbb{R}^n$, we denote its $i$-th coordinate by $a_i$; we use $\|a\|$ to denote its 2-norm (Euclidean norm) and $\|a\|_p$ to denote its $p$-norm for $p \geq 1$. Given two vectors $a, b \in \mathbb{R}^n$, we use $a \cdot b$ to denote their inner product. We denote by $\mathbb{E}[\cdot]$ the expectation with respect to the underlying probability space.

## 2 ALGORITHMS

In this section, we present the main algorithms proposed in this paper: PoweredSGD and PoweredS-GDM. PoweredSGD combines the Powerball function technique with stochastic gradient descent, and PoweredSGDM is an extension of PoweredSGD to include a momentum term. We shall prove in Section 3 that both methods converge and attain at least the best known rates of convergence for SGD and SGDM on non-convex functions, and demonstrate in Section 4 the advantages of using PoweredSGD and PoweredSGDM compared to other popular stochastic optimizers for train deep networks.

### 2.1 POWEREDSGD

Train a DNN with $n$ free parameters can be formulated as an unconstrained optimization problem

$$\min_{x \in \mathbb{R}^n} f(x), \tag{1}$$

where $f(\cdot) : \mathbb{R}^n \to \mathbb{R}$ is a function bounded from below. SGD proved itself an efficient and effective solution for high-dimensional optimization problems. It optimizes $f$ by iteratively updating the parameter vector $x_t \in \mathbb{R}^n$ at step $t$, in the opposite direction of a stochastic gradient $g(x_t, \xi_t)$ (where $\xi_t$ denotes a random variable), which is calculated on $t$-th mini-batch of train dataset. The update rule of SGD for solving problem (1) is

$$x_{t+1} = x_t - \alpha_t g(x_t, \xi_t), \tag{2}$$

starting from an arbitrary initial point $x_1$, where $\alpha_t$ is known as the learning rate at step $t$. In the rest of the article, let $g_t = g(x_t, \xi_t)$ for the sake of notation. We then introduce a nonlinear transformation $\sigma_\gamma(z) = \text{sign}(z)|z|^\gamma$ named as the Powerball function where $\text{sign}(z)$ returns the sign of $z$, or 0 if $z = 0$. For any vector $z = (z_1, \ldots, z_n)^\top$, the Powerball function $\sigma_\gamma(z)$ is applied to all elements of $z$. A parameter $\gamma \in \mathbb{R}$ is introduced to adjust the mechanism and intensity of the Powerball function.

Applying the Powerball function to the stochastic gradient term in the update rule (2) gives the proposed PoweredSGD algorithm:

$$x_{t+1} = x_t - \alpha_t \sigma_\gamma(g_t), \tag{3}$$

where $\gamma \in [0, 1]$ is an additional parameter. Clearly, when $\gamma = 1$, we obtain the vanilla SGD (2). The detailed pseudo-code of the proposed PoweredSGD is presented in Algorithm 1.

---

**Algorithm 1** PowerSGD

1: **Input:** $x_1, \{\alpha_t\}_{t=1}^T, \gamma \in [0, 1]$
2: **for** $t = 1$ **to** $T$ **do**
3:     $g_t = g(x_t, \xi_t)$
4:     $x_{t+1} = x_t - \alpha_t \sigma_\gamma(g_t)$
5: **end for**

---

**Algorithm 2** PowerSGDM

1: **Input:** $x_1, v_1, \{\alpha_t\}_{t=1}^T, \beta \in (0, 1), \gamma \in [0, 1]$
2: **for** $t = 1$ **to** $T$ **do**
3:     $v_{t+1} = \beta v_t - \alpha_t \sigma_\gamma(g_t)$
4:     $x_{t+1} = x_t + v_{t+1}$
5: **end for**

---

### 2.2 POWEREDSGDM

The momentum trick inspired by physical processes Polyak (1964); Nesterov (1983) has been successfully combined with SGD to give SGDM, which almost always gives better convergence rates on train deep networks. We hereby follow this line to propose the PoweredSGD with Momentum (PoweredSGDM), whose update rule is

$$\begin{cases} v_{t+1} = \beta v_t - \alpha_t \sigma(g_t), \\ x_{t+1} = x_t + v_{t+1}. \end{cases} \tag{4}$$

Clearly, when $\beta = 0$, PowerSDGM (4) reduces to PoweredSGD (3). Pseudo-code of the proposed PoweredSGDM is detailed in Algorithm 2.

## 3 CONVERGENCE ANALYSIS

In this section, we present convergence results of PoweredGD and PoweredSGDM in the non-convex setting. We start with some standard technical assumptions. First, we assume that the gradient of the objective function $f$ is $L$-Lipschitz.

**Assumption 3.1** *There exists some $L > 0$ such that $|\nabla f(x) - \nabla f(y)| \leq L\|x - y\|$, for all $x, y \in \mathbb{R}^n$.*

We then assume that a stochastic first-order black-box oracle is accessible as a noisy estimate of the gradient of $f$ at any point $x \in \mathbb{R}^n$, and the variance of the noise is bounded.

**Assumption 3.2** *The stochastic gradient oracle gives independent and unbiased estimate of the gradient and satisfies:*

$$\mathbb{E}[g(x,\xi)] = \nabla f(x), \quad \mathbb{E}[\|g(x,\xi) - \nabla f(x)\|^2] \leq \hat{\sigma}^2, \quad \forall x \in \mathbb{R}^n, \tag{5}$$

*where $\hat{\sigma} \geq 0$ is a constant.*

We will be working with a mini-batch size in the proposed PoweredSGD and PoweredSGDM. Let $n_t$ be the mini-batch size at the $t$-th iteration and the corresponding mini-batch stochastic gradient be given by the average of $n_t$ calls to the above oracle. Then by Assumption 3.2 we can show that $\mathbb{E}[\|g_t - \nabla f(x_t)\|^2] \leq \hat{\sigma}^2/n_t$ for all $t \geq 1$. In other words, we can reduce variance by choosing a larger mini-batch size (see Supplementary Material A.2).

### 3.1 CONVERGENCE ANALYSIS OF POWEREDSGD

We now state the main convergence result for the proposed PoweredSGD.

**Theorem 3.1** *Suppose that Assumptions 3.1 and 3.2 hold. Let $T$ be the number of iterations. PoweredSGD (3) with an adaptive learning rate and mini-batch size $B_t = T$ (independent of a particular step $t$) can lead to*

$$\mathbb{E}\left[\frac{1}{T}\sum_{k=1}^{T}\|g_t\|_{1+\gamma}^2\right] \leq \frac{2\|\mathbf{1}\|_p}{T}\left[\frac{L(f(x_1) - f^\star)}{1 - \varepsilon} + \frac{\hat{\sigma}^2}{2\varepsilon(1 - \varepsilon)}\right],$$

*where $\varepsilon \in (0, 1)$, $p = \frac{1+\gamma}{1-\gamma}$ for any $\gamma \in [0, 1)$ and $p = \infty$ for $\gamma = 1$.*

The proof of Theorem 3.1 can be found in the Supplementary Material A.2.

**Remark 3.1** *The proposed PoweredSGD and PoweredSGDM have the potential to outperform popular stochastic optimizers by allowing the additional parameter $\gamma$ that can be tuned for different training cases, and they always reduce to other optimizers when setting $\gamma = 1$.*

**Remark 3.2** *We leave $\varepsilon \in (0, 1)$ to be a free parameter in the bound to provide trade-offs between bounds given by the curvature $L$ and stochasticity $\hat{\sigma}$. If $\hat{\sigma} = 0$, we can choose $\varepsilon \to 0$ and recover the convergence bound for PoweredGD (see Supplementary Material A.1).*

**Remark 3.3** *The above theorem provides a sharp estimate of the convergence of PoweredSGD in the following sense. When $\gamma = 1$, the convergence bound reduces to the best known convergence rate for SGD. Note that, because of the choice of batch size, it requires $T^2$ gradient evaluations in $T$ iterations. So the convergence rate is effectively $O(1/\sqrt{T})$. This is the best known rate of convergence for SGD Ge et al. (2015). When $\hat{\sigma} = 0$ (i.e., exact gradients are used and $B_t = 1$), PoweredSGD can attain convergence in the order $O(1/T)$, which is consistent with the convergence rate of gradient descent.*

### 3.2 CONVERGENCE ANALYSIS OF POWEREDSGDM

We now present convergence analysis for PoweredSGDM. The proof is again included in the Supplementary Material B.2 due to the space limit.

**Theorem 3.2** *Suppose that Assumptions 3.1 and 3.2 hold. Let $T$ be the number of iterations. For any $\beta \in [0,1)$, PoweredSGDM (4) with an adaptive learning rate and mini-batch size $B_t = T$ (independent of a particular step $t$) can lead to*

$$\mathbb{E}\left[\frac{1}{T}\sum_{t=1}^{T}\|g_t\|_{1+\gamma}^2\right] \leq \frac{2\|\mathbf{1}\|_p}{T}\left[\frac{L(f(x_1)-f^\star)}{1-\varepsilon}\frac{1+\beta}{1-\beta}+\frac{\hat{\sigma}^2}{2\varepsilon(1-\varepsilon)}\right],$$

*where $\varepsilon \in (0,1)$, $p = \frac{1+\gamma}{1-\gamma}$ for any $\gamma \in [0,1)$ and $p = \infty$ for $\gamma = 1$.*

**Remark 3.4** *Convergence analysis of stochastic momentum methods for non-convex optimization is an important but under-explored topic. While our results on convergence analysis do not improve the rate of convergence for stochastic momentum methods in a non-convex setting, it does match the currently best known rate of convergence (Yan et al., 2018; Bernstein et al., 2018) in special cases ($\gamma = 0, 1$) and offers very concise upper bounds in terms of the constants. The upper bound continuously interpolates the convergence rate for $\gamma$ varying in $[0,1]$ and $\beta$ varying in $[0,1)$. The key technical result that made the results of Theorems 3.1 and 3.2 possible is Lemma B.1 in the Supplementary Material, which provide a tight estimate of accumulated momentum terms. We also note that the convergence rates for $\gamma \in (0,1)$ are entirely new and not reported elsewhere before. Even for the special case of $\gamma = 0, 1$, our proof differs from that of (Yan et al., 2018; Bernstein et al., 2018) and seems more transparent.*

**Remark 3.5** *A large mini-batch ($B_t = T$) is assumed for the convergence results to hold. This is consistent with the convergence analysis in Bernstein et al. (2018) for the special case $\gamma = 0$. We assume this because it enables us to put analysis of PoweredGD and PoweredSGD in a unified framework so that we can obtain tighter bounds. In the stochastic setting, similar to Remark 3.3, we note that our proof requires $T^2$ gradient calls in $T$ iterations and hence the effective convergence rate is $O(1/\sqrt{T})$, which is consistent with the known rate of convergence for SGD (Ge et al., 2015).*

## 4 EXPERIMENTS

The propose of this section is to demonstrate the efficiency and effectiveness of the proposed PoweredSGD and PoweredSGDM algorithms. We conduct experiments of different model architectures on datasets in comparison with widely used optimization methods including the non-adaptive method SGDM and three popular adaptive methods: AdaGrad, RMSprop and Adam. This section is mainly composed of two parts: **(1)** the convergence and generalization experiments and **(2)** the Powerball feature experiments. The setup for each experiment is detailed in Table 1[1]. In the first part, we present empirical study of different deep neural network architectures to see how the proposed methods behave in terms of convergence speed and generalization. In the second part, the experiments are conducted to explore the potential features of PoweredSGD and PoweredSGDM.

To ensure stability and reproducibility, we conduct each experiment at least 5 times from randomly initializations and the average results are shown. The settings of hyper-parameters of a specific optimization method that can achieve the best performance on the test set are chosen for comparisons. When two settings achieve similar test performance, the setting which converges faster is adopted.

We can have the following findings from our experiments: **(1)** The proposed PoweredSGD and PoweredSGDM methods exhibit better convergence rate than other adaptive methods such as Adam and RMSprop. **(2)** Our proposed methods achieve better generalization performance than adaptive methods although slightly worse than SGDM. **(3)** The Powerball function can help relieve the gradient vanishing phenomenon.

---

[1]Architectures in generalization and convergence experiments can be found at the following links: **(1)** ResNet-50 and DenseNet-121 on CIFAR-10: `https://github.com/kuangliu/pytorch-cifar`; **(2)** ResNext-29 and WideResNet on CIFAR-100: `https://github.com/junyuseu/pytorch-cifar-models`; **(3)** ResNet50 on ImageNet: `https://github.com/pytorch/examples/tree/master/imagenet`

| Experiments | Datasets | Architecture |
|---|---|---|
| Convergence and Generalization Experiments | CIFAR-10 Krizhevsky & Hinton (2009) | ResNet-50 He et al. (2016) DenseNet-121 Huang et al. (2017) |
| | CIFAR-100 Krizhevsky & Hinton (2009) | ResNext-29 (16x64d) Xie et al. (2017) WideResNet (depth=26, k=10) Zagoruyko & Komodakis (2016) |
| | ImageNet Russakovsky et al. (2015) | ResNet-50 He et al. (2016) |
| Powerball Feature Experiments | MNIST LeCun et al. (1998) | 13-Layer Fully-Connected Neural Network |

Table 1: Summaries of the models and datasets in our experiments.

### 4.1 HYPER-PARAMETER TUNING

Since the initial learning rate has a large impact on the performances of optimizers, we implement a logarithmically-spaced grid search strategy around the default learning rate for each optimization method, and leave the other hyper-parameters to their default settings.

**SGDM:** The default learning rate for SGDM is 0.01. We tune the learning rate on a logarithmic scale from $\{1, 0.1, 0.01, 0.001, 0.0001\}$. The momentum value in all experiments is set to default value 0.9.

**PoweredSGD, PoweredSGDM:** The learning rates for PoweredSGD and PoweredSGDM are chosen from the same range $\{1, 0.1, 0.01, 0.001, 0.0001\}$ as SGDM. The momentum value for PoweredS-GDM is also 0.9. Note that $\gamma = 1$ in Powerball function corresponds to the SGD or SGDM. Based on extensive experiments, we empirically tune $\gamma$ from $\{0.5, 0.6, 0.7, 0.8, 0.9\}$.

**AdaGrad:** The learning rates for AdaGrad are $\{1e\text{-}1, 5e\text{-}2, 1e\text{-}2, 5e\text{-}3, 1e\text{-}3\}$ and we choose 0 for the initial accumulator value.

**RMSprop, Adam:** Both have the default learning rate $1e\text{-}3$ and their learning rates are searched from $\{1e\text{-}2, 5e\text{-}3, 1e\text{-}3, 5e\text{-}4, 1e\text{-}4\}$. The parameters $\beta_1$, $\beta_2$ and the perturbation value $\varepsilon$ are set to default.

As previous findings Wilson et al. (2017) show, adaptive methods generalize worse than non-adaptive methods and carefully tuning the initial learning rate yields significant improvements for them. To better compare with adaptive methods, once we have found the value that was best performing in adaptive methods, we would try the learning rate between the best learning rate and its closest neighbor. For example, if we tried learning rates $\{1e\text{-}2, 5e\text{-}3, 1e\text{-}3, 5e\text{-}4, 1e\text{-}4\}$ and $1e\text{-}4$ was best performing, we would try the learning rate $2e\text{-}4$ to see if performance was improved. We iteratively update the learning rate until performance could not be improved any more. For all experiments, we used a mini-batch size of 128.

### 4.2 EXPERIMENTS: CONVERGENCE AND GENERALIZATION

Fig. 1 shows the learning curves of three experiments we have conducted to observe the performance of PoweredSGD and PoweredSGDM in comparison with other widely-used optimization methods.

**ResNet-50 on CIFAR-10:** We trained a ResNet-50 model on CIFAR-10 and our results are shown in Fig. 1(a) and Fig. 1(b). We ran each experiment for a fixed budget of 160 epochs and reduced the learning rate by a factor of 10 after every 60 epochs Wilson et al. (2017).

As the figure shows, the adaptive methods converged fast and appeared to be performing better than the non-adaptive method SGDM as expected. For PoweredSGD and PoweredSGDM, we observed the same tendency in convergence rate as AdaGrad, which outperformed Adam and RMSprop. For the test performance, adaptive methods and our proposed methods still outperform SGDM in the early stage. SGDM achieved a final best overall test accuracy of 94.75%. The PoweredSGD and PoweredSGDM achieved test accuracies of 94.17% and 94.13% respectively, which are slightly worse than SGDM. The best adaptive method, Adam, achieved a test accuracy of 93.38%.

**WideResNet on CIFAR-100:** Next, we conducted experiments on the CIFAR-100 dataset using WideResNet model. The fixed budget here is 120 epochs and the learning rate reduces by a factor of 10 after every 60 epochs. The results are shown in Fig. 1(e) and Fig. 1(f).

The performance of the PoweredSGD and PoweredSGDM are still promising in both the train set and test set. PoweredSGD, PoweredSGDM and AdaGrad had the fastest initial progress. In the test set, PoweredSGD and PoweredSGDM had much better test accuracy than all other adaptive methods.

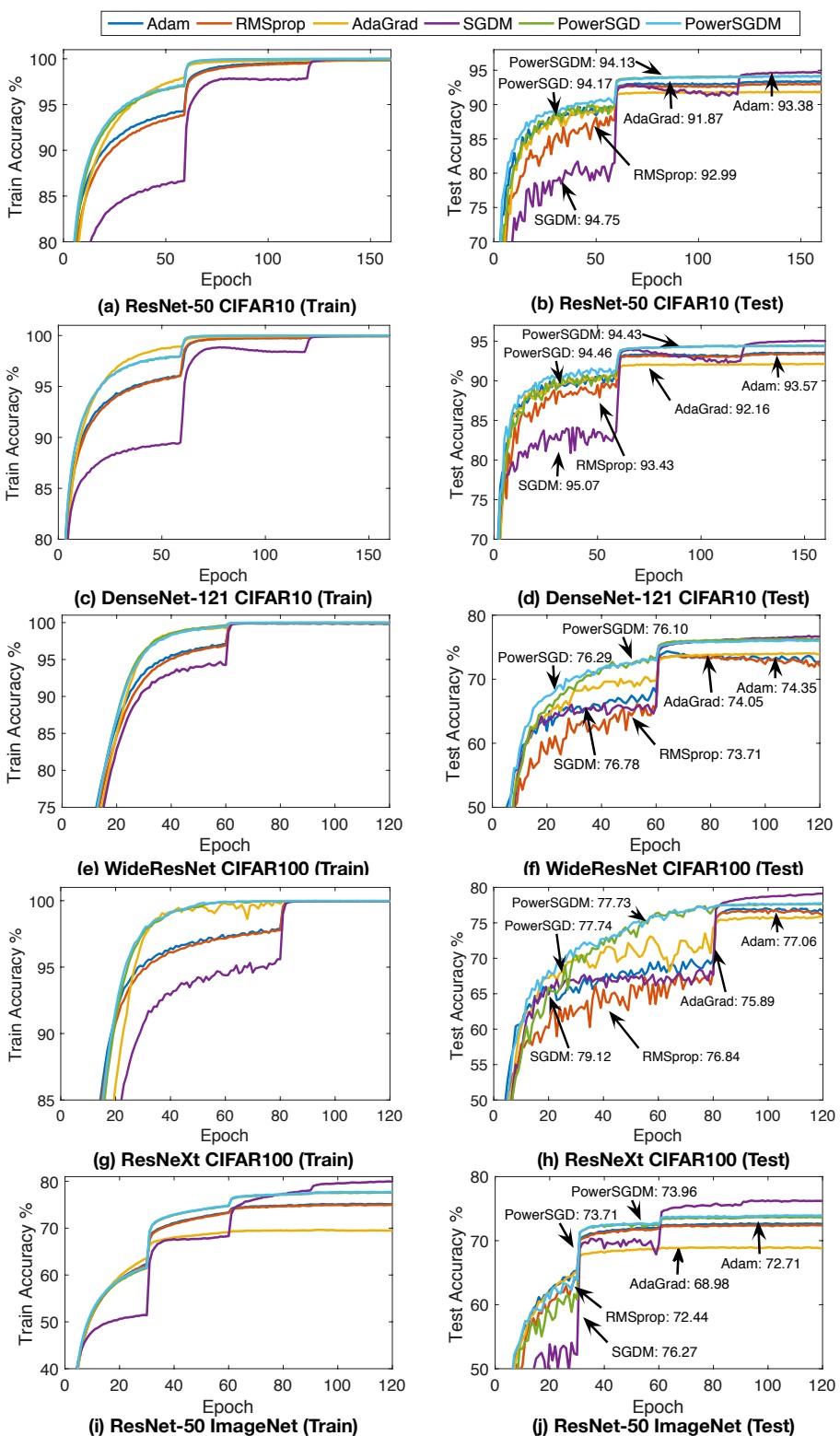

Figure 1: Train and test accuracy for different models and datasets. The annotations indicate the best overall test accuracy for each optimization method. The $\gamma$ values in five experiments are $0.8, 0.8, 0.8, 0.7, 0.8$ for PoweredSGD and $0.8, 0.8, 0.8, 0.8, 0.7$ for PoweredSGDM, respectively.

SGDM surpassed PoweredSGD and PoweredSGDM by epoch 60 when the learning rate decayed. SGDM had the best test accuracy of 76.78%, while PoweredSGD and PoweredSGDM achieved accuracies of 76.29% and 76.10%, respectively, with approximately 0.5% gap in test performance compared with SGDM. The best adaptive methods, still Adam, achieved test accuracy of 74.35% and had much larger gap of 2.5% in performance compared to SGDM.

**ResNet-50 on ImageNet:** Finally, we conducted experiments on the ImageNet dataset using ResNet-50 model. The fixed budget here is 120 epochs and the learning rate reduces by a factor of 10 after every 30 epochs. The results are shown in Fig. 1(i) and Fig. 1(j). We observed that PoweredSGD and PoweredSGDM gave better convergence rates than adaptive methods while AdaGrad quickly plateaus due to too many parameter updates. For test set, we can notice that although SGDM achieved the best test accuracy of 76.27%, PoweredSGD and PoweredSGDM gave the results of 73.71% and 73.96%, which were better than those of adaptive methods.

**Additional experiments (DenseNet-121 on CIFAR-10 and ResNeXt on CIFAR100)** are shown in Fig. 1(c)(d)(g)(h). We observed similar results as in the other experiments.

### 4.3 EXPERIMENTS: FEATURES OF POWEREDSGD

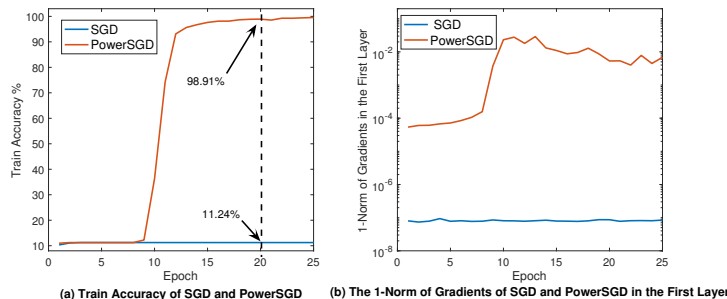

(a) Train Accuracy of SGD and PowerSGD    (b) The 1-Norm of Gradients of SGD and PowerSGD in the First Layer

Figure 2: (a) Train accuracy comparison between SGD (learning rate $= 0.1$) and PoweredSGD (learning rate $= 0.1$, $\gamma = 0.4$) on the 13-layer fully-connected neural network. The arrows annotate the accuracy values of both methods after 20 epochs. (b) The 1-norm of stochastic gradients of SGD and PoweredSGD in the first layer of the fully-connected neural network at every epoch.

**Gradient Vanishing:** In deep learning, the phenomenon of gradient vanishing poses difficulties in training very deep neural networks by SGD. During the training process, the stochastic gradients in early layers can be extremely small due to the chain rule, and this can even completely stop the networks from being trained. Our proposed PoweredSGD method can relieve the phenomenon of gradient vanishing by effectively rescaling the stochastic gradient vectors.

To validate this, we conduct experiments on the MNIST dataset by using a 13-layer fully-connected neural network with ReLU activation functions. The SGD and proposed PoweredSGD are compared in terms of train accuracy and 1-norm of gradient vector. As can be observed in Fig. 2, SGD completely cannot train such a deep network and its train accuracy remained below 11.24%. By contrast, the train accuracy of PoweredSGD grows quickly to 98.91% after few epochs. We further compare the 1-norm of stochastic gradient vector in the first layer for both SGD and PoweredSGD. It is clear that the Powerball function amplifies the stochastic gradients and helps relieve the gradient vanishing phenomenon. We have included more experimental results on gradient vanishing in the Supplementary Material C.

## 5 CONCLUSION

In this paper, a Powerball function is introduced as a basic technique to improve SGD and SGDM for deep neural network training. Their convergence rates have been proved in the non-convex optimization settings. We discussed the choice of an important hyper-parameter $\gamma$ in the Supplementary Material E from an empirical point of view. The experiments on different neural network models and benchmark datasets have demonstrated that the proposed PoweredSGD and PoweredSGDM empirically obtain faster training speed than adaptive gradient methods and good if not better generalization ability compared with SGD. We also show that PoweredSGD can help alleviate gradient vanishing.

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

SUPPLEMENTARY MATERIAL

## A  CONVERGENCE ANALYSIS OF POWEREDSGD AND POWEREDSGDM

### A.1  CONVERGENCE OF POWEREDGD

The authors of Yuan et al. (2016) proposed the so-called Powerball accelerated gradient descent algorithm, which was updated as follows,

$$x_{t+1} = x_t - \alpha_t \sigma(\nabla f(x_t)). \tag{6}$$

The authors of Yuan et al. (2016) then provided insights from finite convergence properties of ODEs models for Powerball variants of gradient descent algorithms and presented empirical observations of convergence acceleration of Powerball methods. Nonetheless, they did not provide any convergence results for their algorithms. In fact, they highlighted the convergence analysis of their algorithms as open theoretical questions. In this paper, we are going to analyze stochastic variants of Powerball methods in the non-convex setting. We start with the analysis of PoweredGD (powered gradient descent) in (6).

**Theorem A.1** *Suppose that Assumption 3.1 holds. The PoweredGD scheme (6) can lead to*

$$\frac{1}{T}\sum_{t=1}^{T}\|\nabla f(x_t))\|_{1+\gamma}^2 \le \frac{2L\|\mathbf{1}\|_p}{T}(f(x_1) - f^\star),$$

*where $T$ is the number of iterations and $p = \frac{1+\gamma}{1-\gamma}$ for any $\gamma \in [0,1)$ and $p = \infty$ for $\gamma = 1$.*

*Proof:* Denote by $x^\star$ the minimizer and $f^\star = f(x^\star)$. Then, by the $L$-Lipschitz continuity of $\nabla f$ and (6),

$$
\begin{aligned}
f(x_{t+1}) &\le f(x_t) + \nabla f(x_t) \cdot (x_{t+1} - x_t) + \frac{L}{2}\|x_{t+1} - x_t\|^2 \\
&= f(x_t) - \alpha_t \nabla f(x_t) \cdot \sigma(\nabla f(x_t)) + \frac{L}{2}\alpha_t^2\|\sigma(\nabla f(x_t))\|^2. \tag{7}
\end{aligned}
$$

Let $\alpha_t = \frac{\nabla f(x_t) \cdot \sigma(\nabla f(x_t))}{L\|\sigma(\nabla f(x_t))\|^2} > 0$ (this holds if $x_t \ne x^\star$). Then

$$f(x_{t+1}) \le f(x_t) - \frac{(\nabla f(x_t) \cdot \sigma(\nabla f(x_t)))^2}{2L\|\sigma(\nabla f(x_t))\|^2}. \tag{8}$$

By Hölder's inequality, for $\gamma \in (0,1)$ and with $p = \frac{1+\gamma}{1-\gamma}$ and $q = \frac{1+\gamma}{2\gamma}$, we have

$$
\begin{aligned}
\|\sigma(\nabla f(x_t))\|^2 &= \sum_{i=1}^{n}|(\nabla f)_i(x_t)|^{2\gamma} \le (\sum_{i=1}^{n}1^p)^{\frac{1}{p}}(\sum_{i=1}^{n}(|(\nabla f)_i(x_t)|^{2\gamma})^q)^{\frac{1}{q}} \\
&\le \|\mathbf{1}\|_p(\sum_{i=1}^{n}|(\nabla f)_i(x_t)|^{1+\gamma})^{\frac{2\gamma}{1+\gamma}}.
\end{aligned}
$$

It follows that

$$
\begin{aligned}
f(x_{t+1}) &\le f(x_t) - \frac{(\nabla f(x_t) \cdot \sigma(\nabla f(x_t)))^2}{2L\|\sigma(\nabla f(x_t))\|^2} \\
&\le f(x_t) - \frac{(\sum_{i=1}^{n}|(\nabla f)_i(x_t)|^{1+\gamma})^2}{2L\|\mathbf{1}\|_p(\sum_{i=1}^{n}|(\nabla f)_i(x_t)|^{1+\gamma})^{\frac{2\gamma}{1+\gamma}}} \\
&= f(x_t) - \frac{1}{2L\|\mathbf{1}\|_p}(\sum_{i=1}^{n}|(\nabla f)_i(x_t)|^{1+\gamma})^{\frac{2}{1+\gamma}} \\
&= f(x_t) - \frac{1}{2L\|\mathbf{1}\|_p}\|\nabla f(x_t)\|_{1+\gamma}^2,
\end{aligned}
$$

which, by a telescoping sum, gives

$$\frac{1}{T}\sum_{t=1}^{T}\|\nabla f(x_t)\|_{1+\gamma}^2 \le \frac{2L\|\mathbf{1}\|_p}{T}(f(x_1) - f(x_{T+1})) \le \frac{2L\|\mathbf{1}\|_p}{T}(f(x_1) - f^\star),$$

where $\mathbf{1}$ is vector with entries all given by 1. It is easy to see that the estimate is also valid for $\gamma = 1$ with $p = \infty$ and for $\gamma = 0$. The proof is complete. ∎

## A.2 CONVERGENCE OF POWEREDSGD (PROOF OF THEOREM 3.1)

To analyze the convergence of PoweredSGD, we need some preliminary on the relation between mini-batch size and variance reduction of SGD.

### VARIANCE REDUCTION BY A LARGER MINI-BATCH SIZE

Let $\nabla f(x)$ be the gradient of $f$ at $x \in \mathbb{R}^n$. Suppose that we use the average of $m$ calls to the stochastic gradient oracle, denoted by $g(x, \xi_i)$ $(i = 1, \cdots, m)$, to estimate $\nabla f(x)$. By Assumption 3.2, we have

$$\mathbb{E}\left[\left\|\frac{\sum_{i=1}^m g(x, \xi_i)}{m} - \nabla f(x)\right\|^2\right] = \frac{1}{m}\mathbb{E}\left[\frac{1}{m}\left\|\sum_{i=1}^m [g(x, \xi_i) - \nabla f(x)]\right\|^2\right]$$

$$= \frac{1}{m}\mathbb{E}\left[\frac{1}{m}\sum_{i=1}^m \|g(x, \xi_i) - \nabla f(x)\|^2\right] = \frac{1}{m}\frac{1}{m}(m\hat{\sigma}^2) = \frac{\hat{\sigma}^2}{m},$$

where in the second equality we used the fact that $g(x, \xi_i)$ $(i = 1, \cdots, m)$ are drawn independently and all give unbiased estimate of $\nabla f(x)$ (provided by Assumption 3.2).

Now we are ready to present the proof of Theorem 3.1.

### PROOF OF THEOREM 3.1

*Proof:* By the $L$-Lipschitz continuity of $\nabla f$ and (6),

$$
\begin{aligned}
f(x_{t+1}) &\leq f(x_t) + \nabla f(x_t) \cdot (x_{t+1} - x_t) + \frac{L}{2}\|x_{t+1} - x_t\|^2 \\
&= f(x_t) - \alpha_t \nabla f(x_t) \cdot \sigma(g_t) + \frac{L}{2}\alpha_t^2\|\sigma(g_t)\|^2 \\
&= f(x_t) - \alpha_t g_t \cdot \sigma(g_t) + \frac{L}{2}\alpha_t^2\|\sigma(g_t)\|^2 + \alpha_t(g_t - \nabla f(x_t)) \cdot \sigma(g_t).
\end{aligned}
$$

Let $\alpha_t = \frac{g_t \cdot \sigma(g_t)}{L\|\sigma(g_t)\|^2} > 0$. Then

$$f(x_{t+1}) \leq f(x_t) - \frac{1}{2L}\frac{(g_t \cdot \sigma(g_t))^2}{\|\sigma(g_t)\|^2} + \alpha_t(g_t - \nabla f(x_t)) \cdot \sigma(g_t). \tag{9}$$

Fix any iteration number $T > 1$ and let $\varepsilon \in (0, 1)$ to be chosen. We can estimate

$$
\begin{aligned}
\alpha_t(g_t - \nabla f(x_t)) \cdot \sigma(g_t) &= \frac{g_t \cdot \sigma(g_t)}{L\|\sigma(g_t)\|^2}(g_t - \nabla f(x_t)) \cdot \sigma(g_t) \\
&\leq \frac{|g_t \cdot \sigma(g_t)|}{L\|\sigma(g_t)\|^2}\|g_t - \nabla f(x_t)\|\|\sigma(g_t)\| \\
&= \frac{|g_t \cdot \sigma(g_t)|}{L\|\sigma(g_t)\|}\|g_t - \nabla f(x_t)\| \\
&\leq \frac{1}{2L}\left(\varepsilon\frac{(g_t \cdot \sigma(g_t))^2}{\|\sigma(g_t)\|^2} + \frac{1}{\varepsilon}\|g_t - \nabla f(x_t)\|^2\right),
\end{aligned}
$$

where the last inequality followed from the elementary inequality $2ab \leq \varepsilon a^2 + \frac{1}{\varepsilon}b^2$ for any positive real number $\varepsilon$ and real numbers $a, b$. Substituting this into (9) gives

$$f(x_{t+1}) \leq f(x_t) - \frac{1-\varepsilon}{2L}\frac{(g_t \cdot \sigma(g_t))^2}{\|\sigma(g_t)\|^2} + \frac{1}{2L\varepsilon}\|g_t - \nabla f(x_t)\|^2. \tag{10}$$

By the same argument in the proof for Theorem A.1, we can derive

$$f(x_{t+1}) \leq f(x_t) - \frac{1-\varepsilon}{2L\|\mathbf{1}\|_p}\|g_t\|_{1+\gamma}^2 + \frac{1}{2L\varepsilon}\|g_t - \nabla f(x_t)\|^2.$$

Taking conditional expectation from both sizes gives

$$
\begin{aligned}
\mathbb{E}[f(x_{t+1}) - f(x_t)\,|\,x_t] &\leq -\frac{1-\varepsilon}{2L\|\mathbf{1}\|_p}\mathbb{E}[\|g_t\|_{1+\gamma}^2] + \frac{1}{2L\varepsilon}\sigma_t^2 \\
&\leq -\frac{1-\varepsilon}{2L\|\mathbf{1}\|_p}\mathbb{E}[\|g_t\|_{1+\gamma}^2] + \frac{\hat{\sigma}^2}{2L\varepsilon T},
\end{aligned}
$$

where $\sigma_t^2$ is the variance of the $t$-th stochastic gradient approximation computed using the chosen mini-batch size $B_t = T$, which therefore satisfies $\sigma_t^2 \leq \frac{\hat{\sigma}^2}{T}$. Taking expectation from both sides and performing a telescoping sum give

$$\mathbb{E}\left[\frac{1}{T}\sum_{t=1}^{T}\|g_t\|_{1+\gamma}^2\right] \leq \frac{2L\|\mathbf{1}\|_p}{T(1-\varepsilon)}(f(x_1)-f^\star) + \frac{\|\mathbf{1}\|_p}{T\varepsilon(1-\varepsilon)}\hat{\sigma}^2.$$

The proof is complete. ∎

## B CONVERGENCE ANALYSIS OF POWEREDSGDM

### B.1 CONVERGENCE OF POWEREDGDM

We first analyze the deterministic version of PoweredSGDM (denoted by PoweredGDM). The update rule for PoweredGDM is

$$\begin{cases} v_{t+1} = \beta v_t - \alpha_t \sigma(\nabla f(x_t)), \\ x_{t+1} = x_t + v_{t+1}, \end{cases} \tag{11}$$

where $\beta \in [0,1)$ is a momentum constant and $v_0 = 0$. Clearly, when $\beta = 0$, the scheme also reduces to PoweredGD.

**Theorem B.1** *Suppose that Assumption 3.1 holds. For any $\beta \in [0,1)$, the PoweredGDM scheme (11) with an adaptive learning rate can lead to*

$$\frac{1}{T}\sum_{t=1}^{T}\|\nabla f(x_t)\|_{1+\gamma}^2 \leq \frac{2L\|\mathbf{1}\|_p}{T}\frac{1+\beta}{1-\beta}(f(x_1)-f^\star),$$

*where $T$ is the number of iterations and $p = \frac{1+\gamma}{1-\gamma}$ for any $\gamma \in [0,1)$ and $p = \infty$ for $\gamma = 1$.*

*Proof:* Let $z_t = x_t + \frac{\beta}{1-\beta}v_t$. It can be verified that the PoweredGDM scheme satisfies

$$\begin{cases} z_{t+1} = z_t - \frac{\alpha_t}{1-\beta}\sigma(\nabla f(x_t)), \\ v_{t+1} = \beta v_t - \alpha_t \sigma(\nabla f(x_t)). \end{cases} \tag{12}$$

By the $L$-Lipschitz continuity of $\nabla f$ and (12),

$$\begin{aligned} f(z_{t+1}) &\leq f(z_t) + \nabla f(z_t)\cdot(z_{t+1}-z_t) + \frac{L}{2}\|z_{t+1}-z_t\|^2 \\ &= f(z_t) - \frac{\alpha_t}{1-\beta}\nabla f(z_t)\cdot\sigma(\nabla f(x_t)) + \frac{L}{2}\frac{\alpha_t^2}{(1-\beta)^2}\|\sigma(\nabla f(x_t))\|^2 \\ &= f(z_t) - \frac{\alpha_t}{1-\beta}\nabla f(x_t)\cdot\sigma(\nabla f(x_t)) - \frac{\alpha_t}{1-\beta}(\nabla f(z_t)-\nabla f(x_t))\cdot\sigma(\nabla f(x_t)) \\ &\quad + \frac{L}{2}\frac{\alpha_t^2}{(1-\beta)^2}\|\sigma(\nabla f(x_t))\|^2. \end{aligned} \tag{13}$$

We can estimate

$$-\frac{\alpha_t}{1-\beta}(\nabla f(z_t)-\nabla f(x_t))\cdot\sigma(\nabla f(x_t)) \leq \frac{1}{2(1-\beta)}\left[\varepsilon\|\nabla f(z_t)-\nabla f(x_t)\|^2 + \frac{1}{\varepsilon}\alpha_t^2\|\sigma(\nabla f(x_t))\|^2\right], \tag{14}$$

where $\varepsilon > 0$ is to be chosen. By the $L$-Lipschitz continuity of $\nabla f$,

$$\|\nabla f(z_t)-\nabla f(x_t)\|^2 \leq L^2\|z_t - x_t\|^2 = L^2\|\frac{\beta}{1-\beta}v_t\|^2 = L^2\frac{\beta^2}{(1-\beta)^2}\|v_t\|^2. \tag{15}$$

**Lemma B.1** *For $T \geq 1$, we have*

$$\sum_{t=1}^{T}\|v_t\|^2 \leq \frac{1}{(1-\beta)^2}\sum_{t=1}^{T}\alpha_t^2\|\sigma(\nabla f(x_t))\|^2.$$

*Proof:* It is easy to show by induction that, for $t \geq 1$,

$$v_t = -\sum_{i=1}^{t-1} \beta^{t-i-1} \alpha_i \sigma(\nabla f(x_i)).$$

Indeed, we have $v_1 = 0$ and $v_2 = -\alpha_1 \sigma(\nabla f(x_1))$. Suppose that the above holds for $t \geq 1$. Then

$$v_{t+1} = \beta v_t - \beta \alpha_t \sigma(\nabla f(x_t)) = \beta \left( -\sum_{i=1}^{t-1} \beta^{t-i-1} \alpha_i \sigma(\nabla f(x_i)) \right) - \beta \alpha_t \nabla f(x_t) = -\sum_{i=1}^{t} \beta^{(t+1)-i-1} \alpha_i \sigma(\nabla f(x_i)).$$

Hence

$$\begin{aligned}
\sum_{t=1}^{T} \|v_t\|^2 &= \sum_{t=1}^{T} \left\| -\sum_{i=1}^{t-1} \beta^{t-i-1} \alpha_i \sigma(\nabla f(x_i)) \right\|^2 \\
&\leq \sum_{t=1}^{T} \left( \sum_{i=1}^{t-1} \beta^{t-i-1} \| \alpha_i \sigma(\nabla f(x_i)) \| \right)^2 \\
&\leq \sum_{t=1}^{T} \sum_{i=1}^{t-1} \beta^{t-i-1} \sum_{i=1}^{t-1} \beta^{t-i-1} \| \alpha_i \sigma(\nabla f(x_i)) \|^2 \\
&\leq \frac{1}{1-\beta} \sum_{t=1}^{T} \sum_{i=1}^{t-1} \beta^{t-i-1} \| \alpha_i \sigma(\nabla f(x_i)) \|^2 \\
&= \frac{1}{1-\beta} \sum_{i=1}^{T} \| \alpha_i \sigma(\nabla f(x_i)) \|^2 \sum_{t=i+1}^{T} \beta^{t-i-1} \\
&\leq \frac{1}{(1-\beta)^2} \sum_{t=1}^{T} \alpha_t^2 \| \sigma(\nabla f(x_t)) \|^2.
\end{aligned}$$

$\blacksquare$

By Lemma B.1, inequalities (14), (15), and a telescoping sum on (13), we get

$$f^\star - f(z_1) \leq -\frac{1}{1-\beta} \sum_{t=1}^{T} \alpha_t \nabla f(x_t) \cdot \sigma(\nabla f(x_t))$$
$$+ \left[ \frac{\varepsilon L^2 \beta^2}{2(1-\beta)^5} + \frac{1}{2\varepsilon(1-\beta)} + \frac{L}{2(1-\beta)^2} \right] \sum_{t=1}^{T} \alpha_t^2 \| \sigma(\nabla f(x_t)) \|^2.$$

It is clear that $\varepsilon = \frac{(1-\beta)^2}{L\beta}$ would minimize the bound on the right-hand side (among different choices of $\varepsilon > 0$) and give

$$f^\star - f(z_1) \leq -\frac{1}{1-\beta} \sum_{t=1}^{T} \alpha_t \nabla f(x_t) \cdot \sigma(\nabla f(x_t)) + \left[ \frac{L\beta}{(1-\beta)^3} + \frac{L}{2(1-\beta)^2} \right] \sum_{t=1}^{T} \alpha_t^2 \| \sigma(\nabla f(x_t)) \|^2.$$

For any $\beta \in [0, 1)$, we can choose $\alpha_t = \frac{\nabla f(x_t) \cdot \sigma(\nabla f(x_t))}{L \| \sigma(\nabla f(x_t)) \|^2} \frac{(1-\beta)^2}{1+\beta}$ so that the bound reduces to

$$f^\star - f(z_1) \leq -\frac{1-\beta}{2L(1+\beta)} \sum_{t=1}^{T} \frac{(\nabla f(x_t) \cdot \sigma(\nabla f(x_t)))^2}{\| \sigma(\nabla f(x_t)) \|^2},$$

which immediately gives the bound in the theorem by noting $z_1 = x_1$.

## B.2 CONVERGENCE OF POWEREDSGDM (PROOF OF THEOREM 3.2)

*Proof:* The proof is built on that for Theorem B.1. With $z_t = x_t + \frac{\beta}{1-\beta} v_t$, it can be verified that the PoweredSGDM scheme satisfies

$$\begin{cases} z_{t+1} = z_t - \dfrac{\alpha_t}{1-\beta} \sigma(g_t), \\ v_{t+1} = \beta v_t - \alpha_t \sigma(g_t). \end{cases} \tag{16}$$

By the $L$-Lipschitz continuity of $\nabla f$ and (4),

$$
\begin{aligned}
f(z_{t+1}) \quad &\leq \quad f(z_t) + \nabla f(z_t) \cdot (z_{t+1} - z_t) + \frac{L}{2} \|z_{t+1} - z_t\|^2 \\
&= \quad f(z_t) - \frac{\alpha_t}{1-\beta} \nabla f(z_t) \cdot \sigma(g_t) + \frac{L}{2} \frac{\alpha_t^2}{(1-\beta)^2} \|\sigma(g_t)\|^2 \\
&= \quad f(z_t) - \frac{\alpha_t}{1-\beta} g_t \cdot \sigma(g_t) - \frac{\alpha_t}{1-\beta} (\nabla f(z_t) - \nabla f(x_t)) \cdot \sigma(g_t) \\
&\qquad + \frac{\alpha_t}{1-\beta} (g_t - \nabla f(x_t)) \cdot \sigma(g_t) + \frac{L}{2} \frac{\alpha_t^2}{(1-\beta)^2} \|\sigma(g_t)\|^2.
\end{aligned}
\tag{17}
$$

Similar to the proof of Theorem B.1, we can estimate

$$
-\frac{\alpha_t}{1-\beta} (\nabla f(z_t) - \nabla f(x_t)) \cdot \sigma(g_t) \quad \leq \frac{1}{2(1-\beta)} \left[ \varepsilon_1 \|\nabla f(z_t) - \nabla f(x_t)\|^2 + \frac{1}{\varepsilon_1} \alpha_t^2 \|\sigma(g_t)\|^2 \right],
\tag{18}
$$

where $\varepsilon_1 > 0$ is to be chosen. By the $L$-Lipschitz continuity of $\nabla f$,

$$
\|\nabla f(z_t) - \nabla f(x_t)\|^2 \leq L^2 \|z_t - x_t\|^2 = L^2 \left\| \frac{\beta}{1-\beta} v_t \right\|^2 = L^2 \frac{\beta^2}{(1-\beta)^2} \|v_t\|^2.
\tag{19}
$$

Similar to Lemma B.1, we obtain

$$
\sum_{t=1}^{T} \|v_t\|^2 \leq \frac{1}{(1-\beta)^2} \sum_{t=1}^{T} \alpha_t^2 \|\sigma(g_t)\|^2.
\tag{20}
$$

We can also bound

$$
\frac{\alpha_t}{1-\beta} (g_t - \nabla f(x_t)) \cdot \sigma(g_t) \leq \frac{1}{2} \left[ \frac{1-\beta}{L\varepsilon(1+\beta)} \|g_t - \nabla f(x_t)\|^2 + L\varepsilon \frac{1+\beta}{(1-\beta)^3} \alpha_t^2 \|\sigma(g_t)\|^2 \right],
\tag{21}
$$

where $\varepsilon > 0$. By inequalities (18)-(21), and a telescoping sum on (17), we get

$$
\begin{aligned}
f^\star - f(z_1) \quad &\leq -\frac{1}{1-\beta} \sum_{t=1}^{T} \alpha_t g_t \cdot \sigma(g_t) + \frac{1}{2} \left[ \frac{\varepsilon_1 L^2 \beta^2}{(1-\beta)^5} + \frac{1}{\varepsilon_1(1-\beta)} + \frac{L}{(1-\beta)^2} + \frac{L\varepsilon(1+\beta)}{(1-\beta)^3} \right] \sum_{t=1}^{T} \alpha_t^2 \|\sigma(g_t)\|^2 \\
&\qquad + \frac{1-\beta}{2L\varepsilon(1+\beta)} \sum_{t=1}^{T} \|g_t - \nabla f(x_t)\|^2.
\end{aligned}
$$

Setting $\varepsilon_1 = \frac{(1-\beta)^2}{L\beta}$ and choosing $\alpha_t = \frac{g_t \cdot \sigma(g_t)}{L\|\sigma(g_t)\|^2} \frac{(1-\beta)^2}{1+\beta}$ lead to

$$
f^\star - f(z_1) \quad \leq -\frac{(1-\beta)(1-\varepsilon)}{2L(1+\beta)} \sum_{t=1}^{T} \frac{(g_t \cdot \sigma(g_t))^2}{\|\sigma(g_t)\|^2} + \frac{1-\beta}{2L\varepsilon(1+\beta)} \sum_{t=1}^{T} \|g_t - \nabla f(x_t)\|^2.
$$

which, by taking expectation from both sides and by the same argument in the proof for Theorem A.1, leads to

$$
f^\star - f(z_1) \leq -\frac{(1-\beta)(1-\varepsilon)}{2L\|\mathbf{1}\|_p(1+\beta)} \mathbb{E} \left[ \sum_{t=1}^{T} \|g_t\|_{1+\gamma}^2 \right] + \frac{\hat{\sigma}^2(1-\beta)}{2L\varepsilon(1+\beta)},
$$

which immediately gives the bound in the theorem by noting $z_1 = x_1$. ∎

**Remark B.1** *Clearly, Theorem 3.2 exactly reduces to Theorem B.1 when $\hat{\sigma} = 0$ and $\varepsilon \to 0$. Moreover, when $\beta = 0$, Theorem 3.2 reduces exactly to Theorem 3.1. This in a sense shows that our estimates are sharp.*

### B.3 ESTIMATES OF TRUE GRADIENTS VS ESTIMATES OF STOCHASTIC GRADIENTS

A careful reader will notice that in Theorems 3.1 and 3.2, our estimates of convergence rates for PoweredSGD and PoweredSGDM, respectively, are in terms of the stochastic gradients $g_t$. We now show that this is without loss of generality in view of Assumption 3.2.

When $\gamma = 1$, we have

$$
\mathbb{E}[\|g_t\|_{1+\gamma}^2] = \mathbb{E}[\|g_t\|_2^2] = \mathbb{E}[\|g_t - \nabla f(x_t) + \nabla f(x_t)\|_2^2] = \mathbb{E}[\|g_t - \nabla f(x_t)\|_2^2] + \mathbb{E}[\|\nabla f(x_t)\|_2^2],
$$

where in the last equality, we used Assumption 3.2. This would imply

$$\mathbb{E}[\|\nabla f(x_t)\|_2^2] \leq \mathbb{E}[\|g_t\|_{1+\gamma}^2] - \mathbb{E}[\|g_t - \nabla f(x_t)\|_2^2] \leq \mathbb{E}[\|g_t\|_{1+\gamma}^2].$$

When $\gamma \in [0,1)$, by the equivalence of norm in $\mathbb{R}^n$, there exist positive constants $C_\gamma$ and $D_\gamma$ such that

$$C_\gamma \|x\|_2^2 \leq \|x\|_{1+\gamma}^2 \leq D_\gamma \|x\|_2^2,$$

for all $x \in \mathbb{R}^n$. Hence

$$\begin{aligned}
\mathbb{E}[\|g_t\|_{1+\gamma}^2] &\geq C_\gamma \mathbb{E}[\|g_t\|_2^2] = C_\gamma \mathbb{E}[\|g_t - \nabla f(x_t) + \nabla f(x_t)\|_2^2] \\
&= C_\gamma \mathbb{E}[\|g_t - \nabla f(x_t)\|_2^2] + C_\gamma \mathbb{E}[\|\nabla f(x_t)\|_2^2] \\
&\geq C_\gamma \mathbb{E}[\|g_t - \nabla f(x_t)\|_2^2] + \frac{C_\gamma}{D_\gamma} \mathbb{E}[\|\nabla f(x_t)\|_{1+\gamma}^2],
\end{aligned}$$

which implies that

$$\mathbb{E}[\|\nabla f(x_t)\|_{1+\gamma}^2] \leq \frac{D_\gamma}{C_\gamma} \mathbb{E}[\|g_t\|_{1+\gamma}^2].$$

In other words, the estimates are equivalent (modulo a constant factor). We prefer the versions in Theorems 3.1 and 3.2, because the bounds are more elegant.

## C  POWEREDSGD HELPS ALLEVIATE THE VANISHING GRADIENT PROBLEM

The vanishing gradient problem is quite common when training deep neural networks using gradient-based methods and backpropagation. The gradients can become too small for updating the weight values. Eventually, this may stop the networks from further training. The Powerball function can help amplify the gradients especially when they approach zero. We visualized the amplification effects of Powerball function in Fig. 3. Thus, the attributes of PoweredSGD can help alleviate the vanishing gradient problem to some extent. We investigated the actual performance of PoweredSGD and SGD when dealing with very deep networks.

We trained deep networks on the MNIST dataset using PoweredSGD with $\gamma$ chosen from $\{0, 0.1, 0.2, 0.3, 0.4, 0.5, 0.6, 0.7, 0.8, 0.9, 1.0\}$ and learning rate $\eta$ chosen from $\{1.0, 0.1, 0.01, 0.001\}$. When $\gamma = 1.0$, the PoweredSGD becomes the vanilla SGD. The architecture of network depth which ranges from 12 to 15 with ReLU as the activation function is shown in Table 2. The results are visualized using heatmaps in Fig. 4.

|  | Hidden neurons |
| --- | --- |
| Input layer | $784 \rightarrow 256$ |
| Hidde layers ($\times 10/11/12/13$) | $256 \rightarrow 256$ |
| Output layer | $256 \rightarrow 10$ |

Table 2: The architecture of MLP in vanishing gradient experiments.

As we can observe in the visualisation, when the network depth is more than 13 layers, increasing or decreasing the learning rate of SGD could not solve the vanishing gradient problem. For PoweredSGD, the usage of the Powerball function enables it to amplify the gradients and thus allows to further train deep networks with proper $\gamma$ settings. This confirms our hypothesis that PoweredSGD helps alleviate the vanishing gradient problem to some extent. We also note that, when the network increases to 15 layers, both SGD and PoweredSGD could not train the network further. We speculate that this is due to the ratio of amplified gradients to the original gradients becomes too large (see Fig. 3) and a much smaller learning rate is needed (this is also consistent with the change of theoretical learning rates suggested in the convergence proofs as the gradient size decreases). Since PoweredSGD is essentially a gradient modifier, it would also be interesting to see how to combine it with other techniques for dealing with the vanishing gradient problem. Since PoweredSGD also reduces the gradient when the gradient size is large, it may also help alleviate the exploding gradient problem. This gives another interesting direction for future research.

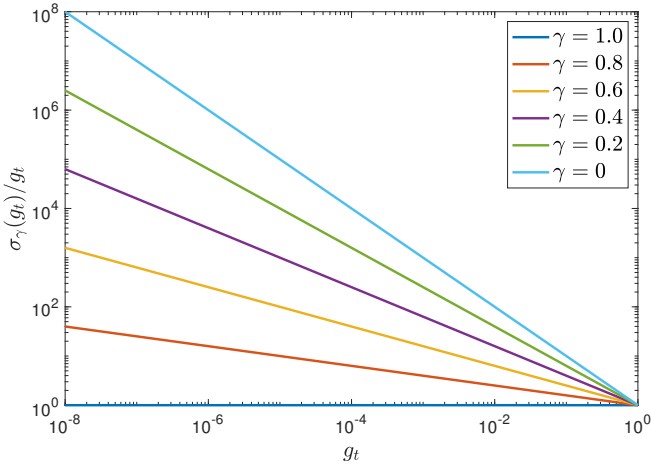

Figure 3: The amplification effects of Powerball function with different $\gamma$ and different gradient sizes. The amplification ratio is larger when the gradient size is closer to 0.

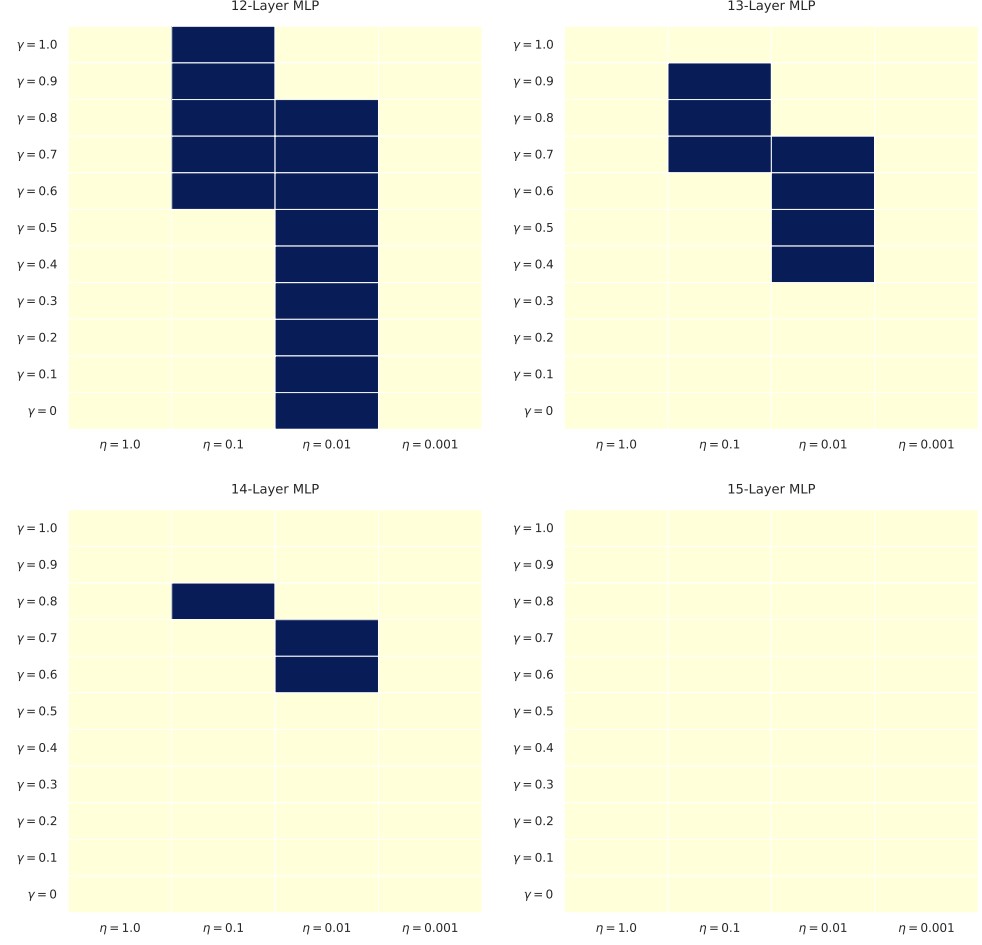

Figure 4: We trained deep networks using PoweredSGD with different hyper-parameter settings. A dark block indicates that PoweredSGD with a specific setting succeeded in training the network within 25 epochs, while a light block indicates that PoweredSGD (or SGD) failed to train. It is clear that by tuning the hyper-parameter $\gamma$, PoweredSGD can train networks deeper than that by SGD.

# D    IMPROVED ROBUSTNESS TO HYPER-PARAMETER SELECTION

The Powerball function is a nonlinear function with a tunable hyper-parameter $\gamma$ applied to gradients, which is introduced to accelerate optimization. To test the robustness of different $\gamma$, we trained ResNet-50 and DenseNet-121 on the CIFAR-10 dataset with PoweredSGD and SGDM. The parameter $\gamma$ is chosen from $\{0, 0.1, 0.2, 0.3, 0.4, 0.5, 0.6, 0.7, 0.8, 0.9, 1.0\}$ and the learning rate is chosen from $\{1.0, 0.1, 0.01, 0.001\}$. The PoweredSGD becomes the vanilla SGD when $\gamma = 1$. The maximum test accuracy is recorded and the results are visualized in Fig. 5.

Although the $\gamma$ that gets the best test accuracy depends on the choice of learning rates, we can observe that $\gamma$ can be selected within a wide range from 0.5 to 1.0 without much loss in test accuracy. Moreover, the Powerball function with a hyper-parameter $\gamma$ could help regularize the test performance while the learning rate decreases. For example, when $\eta = 0.001$ and $\gamma = 0.6$, PoweredSGD get the best test accuracy of 90.06% compared with 79.87% accuracy of SGD.

We also compare the convergence performance of different $\gamma$ choice in Fig. 6. The training loss is recorded when training ResNet-50 on CIFAR-10 dataset. As the initial learning rate decreases, the range from which the hyper-parameter $\gamma$ can be selected to accelerate training becomes wider. As a practical guide, $\gamma = 0.8$ seems a proper setting in most cases. It is again observed that the choice of $\gamma$ in the range of 0.4–0.8 seems to provide improved robustness to the change of learning rates.

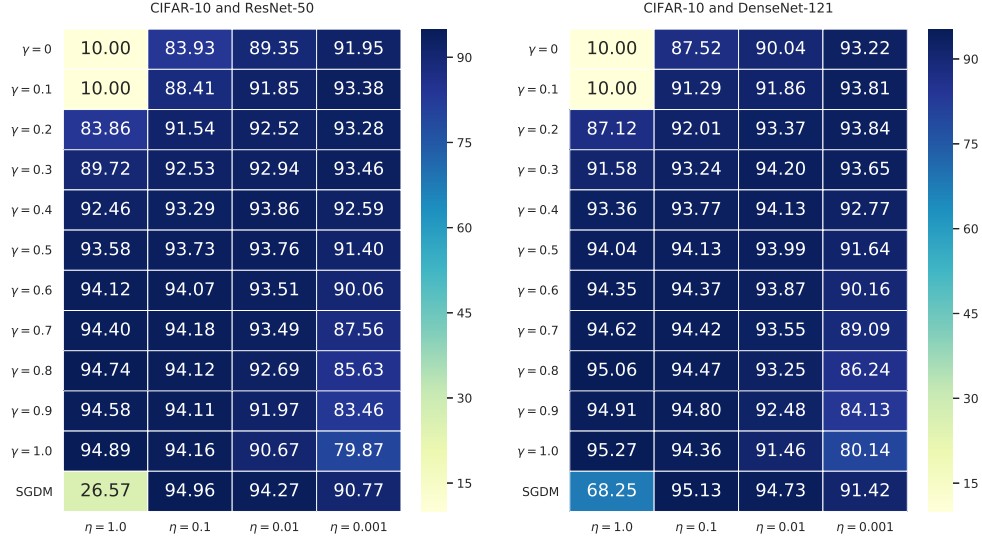

Figure 5: Effects of different $\gamma$ on test accuracy. We show the best Top-1 accuracy on CIFAR-10 dataset of ResNet-50 and DenseNet121 trained with PoweredSGD. Although the best choice of $\gamma$ depends on learning rates, the selections can be quite robust considering the test accuracy.

# E    COMBINING POWEREDSGD WITH LEARNING RATE SCHEDULES

In the main part of the paper, we demonstrated through multiple experiments that PoweredSGD can achieve faster initial training. In this section we demonstrate that PoweredSGD as a gradient modifier is orthogonal and complementary to other techniques for improved learning.

The learning rate is the most important hyper-parameter to tune for deep neural networks. Motivated by recent advancement in designing learning rate schedules such as CLR policies (Smith, 2015) and SGDR (Loshchilov & Hutter, 2016), we conducted some preliminary experiments on combining learning rate schedules with PoweredSGD to improve its performance. The results are shown in Fig. 7.

The selected learning rate schedule is warm restarts introduced in (Loshchilov & Hutter, 2016), which reset the learning rate to the initial value after a cycle of decaying the learning rate with a

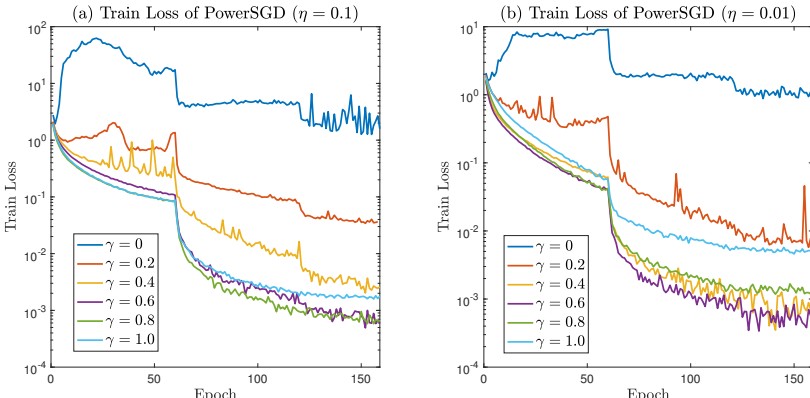

Figure 6: Effects of different $\gamma$ on convergence. We show the best train loss on CIFAR-10 dataset of ResNet-50 trained with PoweredSGD. While the $\gamma$ which achieves the best convergence performance is closely related to the choice of learning rates, a $\gamma$ chosen in the range of 0.4–0.6 seem to provide better robustness to change of learning rates.

cosine annealing for each batch. In Fig. 7, SGD with momentum combined with warm restarts policy is named as SGDR. Similarly, PoweredSGDR indicates PoweredSGD combined with a warm restarts policy. The hyper-parameter setting is $T_0 = 10$ and $T_{mult} = 2$ for warm restarts. We test their performance on CIFAR-10 dataset with ResNet-50.

The results showed that the learning rate policy can improve both the convergence and test accuracy of PoweredSGD. Indeed, PoweredSGDR achieved the lowest training error compared with SGDM and SGDR. The test accuracy for PoweredSGDR was also improved from the 94.12% accuracy of PoweredSGD to 94.64%. The results demonstrate that the nonlinear transformation of gradients given by the Powerball function is orthogonal and complementary to existing methods. As such, its combination with other techniques could potentially further improve the performance.

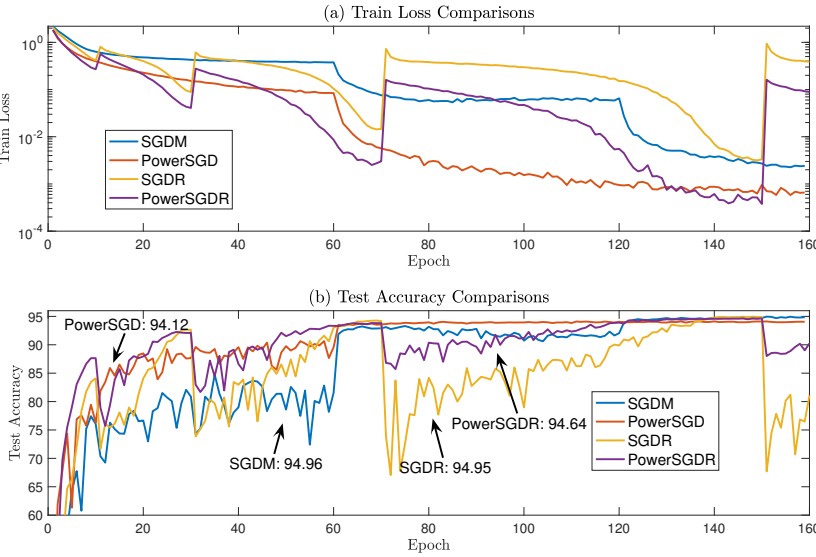

Figure 7: Train loss and test accuracy for SGDR and PoweredSGDR. Learning rate schedules also help accelerate training of PoweredSGD and improve the test performance.

# F  COMPARISONS WITH VANILLA SGD AND SGDM

In this section, we compare PoweredSGD with the vanilla SGD and SGDM to show that how a simple Powerball function can boost the performance. The first four experiments in Section 4.2 are conducted for comparisons. The results are shown in Figure 8 below, in which the hyper-parameters that lead to the best test accuracy are chosen and can be found in Table 3.

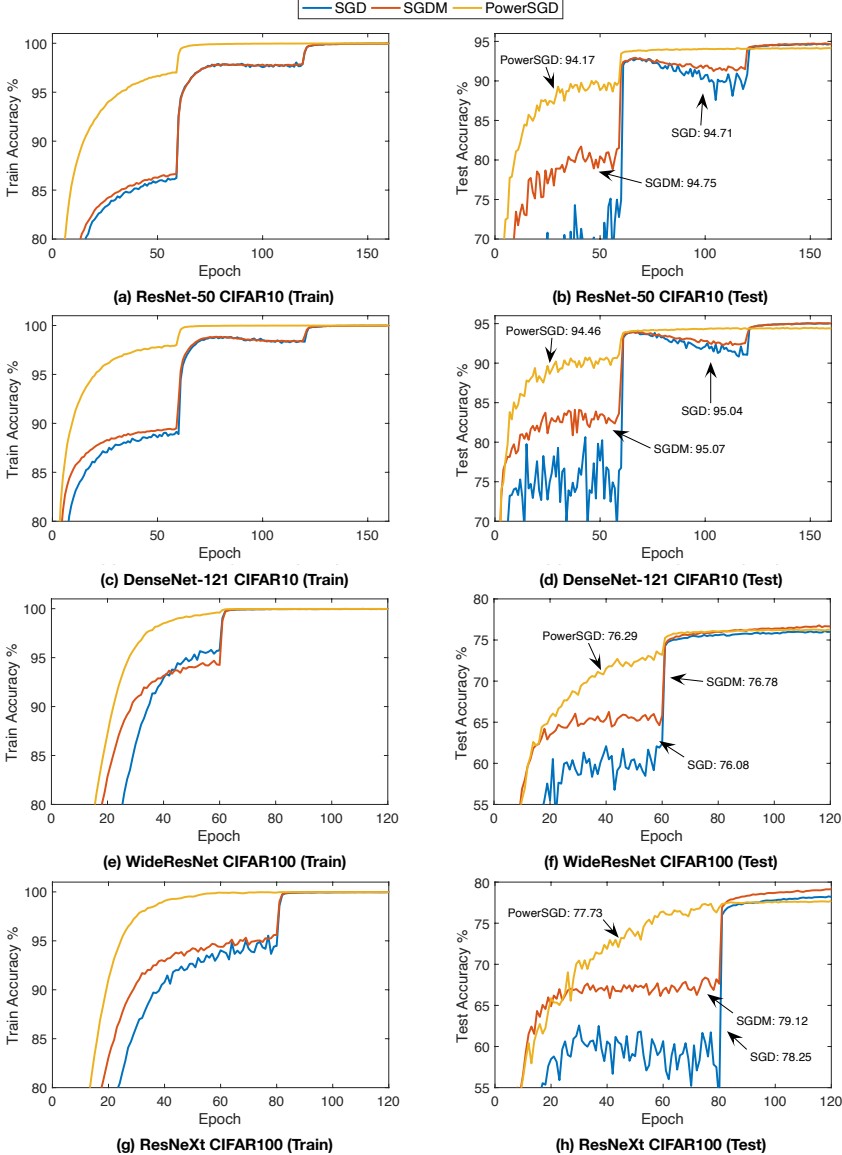

Figure 8: Train and test accuracy for SGD, SGDM and PowerSGD. The annotations indicate the best overall test accuracy for each optimization method.

|  | SGD | SGDM | PowerSGD |
|---|---|---|---|
| ResNet-50 + CIFAR10 | $\eta = 1.0$ | $\eta = 0.1$ | $\eta = 0.1, \gamma = 0.8$ |
| DenseNet-121 + CIFAR10 | $\eta = 0.6$ | $\eta = 0.08$ | $\eta = 0.1, \gamma = 0.8$ |
| WideResNet + CIFAR100 | $\eta = 0.6$ | $\eta = 0.1$ | $\eta = 0.1, \gamma = 0.8$ |
| ResNeXt + CIFAR100 | $\eta = 1.0$ | $\eta = 0.08$ | $\eta = 0.1, \gamma = 0.7$ |

Table 3: Hyper-parameter settings for experiments shown in Figure 8.

