# OpenReview forum: "PowerSGD: Powered Stochastic Gradient Descent Methods for Accelerated Non-Convex Optimization"
_ICLR.cc/2020/Conference — Reject_

### Official Review · AnonReviewer1 · 2019-10-20
**Official Blind Review #1**

**Rating:** 3

**Review:**

This paper investigates an SGD variant (PowerSGD) where the stochastic gradient is raised to a power of $\gamma \in [0,1]$.  The authors introduce PowerSGD and PowerSGD with momentum (PowerSGDM). The theoretical proof of  the convergence is given and experimental results show that the proposed algorithm converges faster than some of the existing popular adaptive SGD techniques.  Intuitively, the proposed PowerSGD can boost the gradient (since $\gamma \in [0,1]$) so it may be helpful for the gradient of the lower layers of a deep network which may be hit by the vanishing gradient issue. This may give rise to a faster convergence.   So overall the idea makes sense but I have the following concerns.

1. The major issue I have with this paper is Theorem 3.1 on the ergodic convergence rate of the proposed PowerSGD.  At the first glance, it is $O(\frac{1}{T})$ which is faster than the conventional SGD convergence rate $O(\frac{1}{\sqrt{T}})$.  But after a closer look, this rate is obtained by a very strong assumption on the batch size $B_{t}=T$.  In other words, when the number of iterations is large, the batch size will be large too.  I consider this assumption unrealistic.  Given that $T$ is typically very large (it is iterations, not epochs),  it will require a huge batch size, probably close to the whole training set. In this case, it is basically a GD, not SGD any more. That's why the rate is $O(\frac{1}{T})$, which is the convergence rate of GD.   I would like to see a convergence proof where the batch size $B_{t}$ is treated as a small constant like other SGD proofs assume.  Actually in the experiments the authors never use an increasing batch size. Instead, a constant batch size 128 is used. Therefore,  the faster convergence demonstrated in the experiments can not be explained by Theorem 3.1 or Theorem 3.2.

2. There are numerous inaccuracies in the proof given the supplementary material.  For instance, in Eq.7,  $\nabla f(x) \sigma(\nabla f(x))$  should be $\nabla f(x)^{T} \sigma(\nabla f(x))$   The random variable $\xi_{t}$ should be a scalar on training samples, not a vector, etc..  The authors should clean it up.

3. It would be helpful to show the $\gamma$ value on each experiment with different tasks. It would be good to know how $\gamma$ varies across tasks.

4. I think in the comparative experiments, the plain SGD should be added as another reference algorithm.

5. The term "PowerSGD" seems to have been used by other papers.


**Experience Assessment:**

I have read many papers in this area.

**Review Assessment: Checking Correctness Of Derivations And Theory:**

I carefully checked the derivations and theory.

**Review Assessment: Checking Correctness Of Experiments:**

I assessed the sensibility of the experiments.

**Review Assessment: Thoroughness In Paper Reading:**

I read the paper at least twice and used my best judgement in assessing the paper.

---

> ### Author Response · Authors · 2019-11-10
> **Response to Reviewer 1**
>
> We thank you for your comments and hope that the following response will address your concerns.
>
> 1. We did stated both in the Theorem statements and Remark 3.4 that the a large batch size $B_t=T$ is used for the convergence proof. This means the effective rate of convergence is $O(1/\sqrt{T})$ as pointed out by the reviewer. This rate matches the currently best known rate of convergence for SGD (see, e.g. Ge et al., COLT'15). We have now made this very clear in both Remarks 3.3 and 3.5. Please see changes highlighted in blue and also our response to Reviewer 2 on novelty of the convergence analysis.
>
> If our response addresses your main concern, we sincerely hope you that you can reconsider your score.
>
> For your other points, we have made the following changes in the paper.
>
> 2. We have checked and fixed a few typos in the paper. Please note that we wrote  $\nabla f(x)\cdot \sigma (\nabla f(x))$ in eq. (7) as a dot product. So it is the same as $\nabla f(x)^{T} \sigma(\nabla f(x))$. This notation was explained in the notation section. If you have any remaining concerns, please let us know.
>
> 3. We have added the values for chosen $\gamma$ in the updated version (see caption of Figure 1).
>
> 4. We plan to add SGD to the experiments, but this may take a while to complete, especially for some of the experiments. We promise to do so in the final version.
>
> 5. We were not aware of this at the time of submission. We have changed this to PoweredSGD. If you have any alternative suggestions, please let us know.
>
> We summarize the main contributions of the paper as follows:
>
> - In the theoretical part, we provided more concise convergence rate analysis for stochastic momentum methods in the non-convex setting. This was made possible by a sharp estimate of the accumulated momentum terms (Lemma B1). We believe this is an important but under-explored topic (Yan et al., 2018).
>
> - In the experimental part, we empirically showed that the proposed optimisation algorithms have potential to solve realistic problems. We are not claiming these variants will outperform all other methods in all training cases, but we sincerely believe that the results are promising. In particular, we have demonstrated their potential benefits of mitigating gradient vanishing and combining other techniques for accelerating optimization.
>
> We do admit the gap between our theoretical analysis and experiments in the sense that the analysis does not account for the initial acceleration observed in many experiments. We think this is a very interesting question for future research and hope that this paper can motivate further research in this area. We agree with your intuition that this may have something to do with $\gamma\in (0,1)$ boosting the gradients.

---

### Official Review · AnonReviewer2 · 2019-10-23
**Official Blind Review #2**

**Rating:** 3

**Review:**

This paper proposes PowerSGD for improving SGD to train deep neural networks. The main idea is to raise the stochastic gradient to a certain power. Convergence analysis and experimental results on CIFAR-10/CIFAR-100/Imagenet and classical CNN architectures are given.

Overall, this is a clearly-written paper with comprehensive experiments. My major concern is whether the results are significant enough to deserve acceptance. The proposed method PowerSGD is an extension of the method in Yuan et al. (extended to handle stochastic gradient and momentum). I am not sure how novel the convergence analysis for PowerSGD is, and it would be nice if the authors could discuss technical challenges they overcome in the introduction.

**Experience Assessment:**

I have read many papers in this area.

**Review Assessment: Checking Correctness Of Derivations And Theory:**

I assessed the sensibility of the derivations and theory.

**Review Assessment: Checking Correctness Of Experiments:**

I assessed the sensibility of the experiments.

**Review Assessment: Thoroughness In Paper Reading:**

I read the paper at least twice and used my best judgement in assessing the paper.

---

> ### Author Response · Authors · 2019-11-10
> **Response to Reviewer 2**
>
> We thank the reviewer for the comments. We justify the novelty and significance of the contributions made by this paper as follows.
>
> 1) Novelty of the convergence analysis: The paper by Yuan et al. did not present proof of convergence in the discrete-time setting. The authors only provided convergence of the ODE models. On the other hand, convergence analysis of momentum methods in non-convex setting is an important but under-explored area  (Yan et al., 2018). In the current paper, the convergence results are proved for non-convex objective functions satisfying mild assumptions. Appropriate use of some sharp estimates allowed us to obtain concise bounds on convergence of the entire class of PoweredSGD methods for $\gamma\in[0,1]$ and the bounds continuously depend on parameters $\gamma$ and $\beta$. In the special cases ($\gamma=0,1$, $\beta=0$), these bounds matches the best known bounds for GD/SGD/SGDM in the non-convex setting. More specifically, we would like to draw the reviewer's attention to the following two papers:
>
> *   [Yan18] Yan, Y., T. Yang, Z. Li, Q. Lin, and Y. Yang. "A unified analysis of stochastic momentum methods for deep learning." In IJCAI International Joint Conference on Artificial Intelligence. 2018.
>
>   *  [Bernstein18] Bernstein, Jeremy, Yu-Xiang Wang, Kamyar Azizzadenesheli, and Animashree Anandkumar. "SIGNSGD: Compressed Optimisation for Non-Convex Problems." In International Conference on Machine Learning, pp. 559-568. 2018. (Theorem 3)
>
> We emphasize that our theoretical analysis leads to significant more concise convergence bounds than those in the above papers. Please take a look at Theorems 1 and 2 in [Yan18] and Theorem 3 in [Bernstein18]. We have highlighted the technical challenges we overcome in order to obtain these results in the updated version (please see Remark 3.3).
>
> 2) Novelty of experiments: The current paper presents substantially more comprehensive experiments for benchmarking the proposed class of optimizers against other popular optimization methods for deep learning tasks. In particular, we highlight the experiments on vanishing gradients and learning rate schedules. This, in addition to the potential to accelerate initial convergence, makes the proposed PoweredSGD methods useful in many potential applications.

---

### Official Review · AnonReviewer3 · 2019-10-24
**Official Blind Review #3**

**Rating:** 8

**Review:**

This paper proposes, analyzes, and empirically evaluates PowerSGD (and a version with momentum), a simple adjustment to standard SGD algorithms that alleviates issues caused by poorly scaled gradients in SGD. The rates in the theoretical analysis are competitive with those for standard SGD, and the empirical results argue that PowerSGD algorithms are competitive with widely used adaptive methods such as Adam and RMSProp, suggesting that PowerSGD may be a useful addition to the armory of adaptive SGD algorithms.

Overall I recommend acceptance of this paper, although I think there may be a couple of places where the authors overclaim a bit on the theoretical side. Specifically:
• The convergence analysis assumes a batch size equal to T, the number of steps of PowerSGD. This implies that the amount of work (in FLOPs) done by the algorithm (at least the version being analyzed) is quadratic in T, which makes the convergence rates a bit misleading. If one reframes the convergence rate in terms of FLOPs U=T^2 instead of iterations, then the convergence rate drops from 1/T to 1/sqrt(U), which undermines the claim in remark 3.4 that this analysis is superior to that of Yan et al. (2018).
• In Remark 3.4.3, the authors claim that another point of difference between their results and Yan et al.'s (2018) is that Yan et al. assume bounded gradients, an assumption that is not satisfied for e.g., mean squared error (MSE). But a very similar assumption is hidden in the bounded-gradient-variance assumption Assumption 3.2; for example, Assumption 3.2 is clearly not satisfied by the least-squares regression problem
min_β (1/N)Σ_n (y_n – x_n • β)^2
with the minibatch gradient estimator computed over randomly chosen minibatches B:
\hat g = (1/|B|) Σ_{n \in B} x_n (y_n – x_n • β).
As the norm of β goes to infinity, so does the expected norm of the error of \hat g. I'm not saying this is a particularly big
deal, just that it's not an improvement over Yan et al.'s result.

That aside, this seems like good work that could have a significant impact on practice.

A couple of other minor points:
• It looks like neither the experiments nor Theorem 3.2 show any benefit to PowerSGDM over PowerSGD. It would be nice to see some discussion (or at least speculation) on why that is.
• Not all of the arrows in Figure 1 are pointing to the right lines.
• In the abstract, it might be good to clarify that the exponentiation is elementwise.

**Experience Assessment:**

I have read many papers in this area.

**Review Assessment: Checking Correctness Of Derivations And Theory:**

I assessed the sensibility of the derivations and theory.

**Review Assessment: Checking Correctness Of Experiments:**

I carefully checked the experiments.

**Review Assessment: Thoroughness In Paper Reading:**

I read the paper at least twice and used my best judgement in assessing the paper.

---

> ### Author Response · Authors · 2019-11-10
> **Response to Reviewer 3**
>
> We thank the reviewer for the positive assessment of our work. We would like start by stating that we did not mean to claim that the rate of convergence proved in this paper is better that than of Yan et al. We have modified the Remarks to clarify the statements. In the stochastic gradient setting, the number of gradient evaluation is indeed $T^2$. This is consistent with the result in Bernstein et al. (2018). The main point we would like to make is that the bounds are very concise and exactly reduce to that of gradient descent/stochastic gradient descent in the special cases. We thank you for pointing out that the bounded variance assumption may also be restrictive and only satisfied on bounded domains. It is nonetheless a standard assumption made in the literature. We have modified Remark 3.4 (and added Remark 3.5) to make this clear in the updated version.
>
> Response to other minor points:
>
> Our convergence analysis is done for non-convex objective functions (similar to that of Yan et al. and Bernstein et al.). In the non-convex setting, to the best of our knowledge, there are no theoretical results that show benefits of momentum methods over SGD. For experiments, we speculate that the reason is that the batch size used is too small for (Powered)SGDM to gain an advantage over (Powered)SGD. We plan to add SGD as a reference algorithm (as suggested by another reviewer). Once the experiments are complete, we should be able to see how SGDM compares with SGD in the experiments. This may take a while for the ImageNet experiments, but we promise to do so in the final version.
>
> We have fixed the other issues you raised in your other minor comments. If you have any further comments, please let us know.

---

### Public Comment · ~Boris_Ginsburg1 · 2019-10-06
**Comparison with adaptive methods for Resnet-50 on  ImageNet**

I would argue that Section 4.2 used wrong adaptive methods for comparison, and this lead to the wrong conclusion: "For test set, we can notice that although SGDM achieved the best test accuracy of 76.27%, PowerSGD and PowerSGDM gave the results of 73.71% and 73.96%, which were better than those of adaptive methods"
There are many adaptive methods which give top1 accuracy similar toSGDM or better.
For example:
1.  AdamW: top-1=76.3% in 100 epochs ( hype-parameters and training details  https://github.com/NVIDIA/OpenSeq2Seq/blob/master/example_configs/image2label/resnet-50v2-adamw.py)
2. LAMB:  top-1 = 76.6 (https://arxiv.org/pdf/1904.00962.pdf)
3. NovoGrad: top-1 = 77% (https://arxiv.org/abs/1905.11286 )

---

> ### Author Response · Authors · 2019-10-10
> **Response**
>
> Thank you for the comment.
>
> While there are numerous variants of adaptive methods available, we focused our comparisons with the most popular ones, namely, Adam, AdaGrad and RMSprop with their standard setup for batchsize, step decay scheme. For example, the batch size is set to be 256, the same as [1,2,3]. The stepsize decay scheme is the same as [2,3].
>
> We also thank you, as the lead author of NovoGrad for bringing your work to our attention. We note that both Lamb [4] and NovoGrad [5] are also under review for ICLR20 (code not available with the submissions, therefore we cannot reproduce and make a fair comparison). It is worthy mentioning that the batch size and decay scheme set in [4] and [5] are different from [1,2,3].
>
> It would be interesting to experiment in future work on how to compare and combine the technique we propose (a nonlinear gradient transformation that is complementary and orthogonal) with other adaptive techniques such as the ones you pointed out.
>
> [1] Kaiming He, Xiangyu Zhang, Shaoqing Ren, and Jian Sun. "Deep residual learning for image recognition." In Proceedings of the IEEE conference on computer vision and pattern recognition, pp. 770-778. 2016.
>
> [2] Gao Huang, Zhuang Liu, Laurens Van Der Maaten, and Kilian Q. Weinberger. "Densely connected convolutional networks." In Proceedings of the IEEE conference on computer vision and pattern recognition, pp. 4700-4708. 2017.
>
> [3] https://github.com/pytorch/examples/tree/master/imagenet
>
> [4] Yang You, Jing Li, Sashank Reddi, Jonathan Hseu, Sanjiv Kumar, Srinadh Bhojanapalli, Xiaodan Song, James Demmel, Kurt Keutzer, Cho-Jui Hsieh. "Large Batch Optimization for Deep Learning: Training BERT in 76 minutes." arXiv preprint arXiv:1904.00962, 2019.
>
> [5] Boris Ginsburg, Patrice Castonguay, Oleksii Hrinchuk, Oleksii Kuchaiev, Vitaly Lavrukhin, Ryan Leary, Jason Li, Huyen Nguyen, Yang Zhang, Jonathan M. Cohen. "Stochastic Gradient Methods with Layer-wise Adaptive Moments for Training of Deep Networks." arXiv preprint arXiv:1905.11286, 2019.

---

### Author Response · Authors · 2019-11-12
**To all reviewers**

We have now included the experiment results for comparisons with the plain SGD as requested by the reviewers. Since Figure.1 is already very crowded, we have included these comparisons in Appendix F. So far the experiments are not complete (due to the time constraints) in the sense that we are missing the experiments on ImageNet and we have only completed 1 run of SGD (whereas all others are averages of 5 runs). We will

- by Nov 15, complete the comparisons with plain SGD for the first 4 sets of experiments;
- by the time of final submission (should the paper be accepted),  comparisons with plain SGD for all 5 experiments.

We thank you again for taking the time to review our paper, and hope that you can comment on the updated version and let us know if you have any remaining concerns. We would like to reiterate the main contributions of the paper as follows:

- In the theoretical part, we provided more concise convergence rate analysis for stochastic momentum methods in the non-convex setting. The analysis for $\gamma\in(0,1)$ is completely new and our proof derived tighter estimates of the accumulated momentum terms, which may be of interest for other uses. The bounds obtained are concise, change continuously w.r.t to the parameters $\gamma\in[0,1]$ and $\beta\in[0,1)$, and match the best known convergence rates in special cases.

- In the experimental part, we empirically showed through a comprehensive set of experiments that PoweredSGD can speed up initial training in many cases, can partially mitigate gradient vanishing, and can be easily combined with other techniques to further accelerate optimization.

The main gap between the theoretical part and experimental part lies in that we are not able to explain the initial acceleration from a theoretical point of view. How to close this gap is of course an interesting and important question. We note however that there are rarely methods that can universally outperform other methods (both theoretically and empirically). We hope this paper can motivate further research on this topic, both in terms of theoretical analysis and practical use of the proposed methods.

---

> ### Author Response · Authors · 2019-11-15
> **update**
>
> We have completed the comparisons with plain SGD for the first 4 sets of experiments. Figure 8 in Appendix F is now updated.
>
> We also removed the changes highlighted in blue. You should be able to see the changes from the original submissions using pdf diff.
>
> We hope that our response has addressed your concerns and look forward to any further comments you may have.

---

### Decision · Program_Chairs · 2019-12-19

**Decision:**

Reject

**Comment:**

After reading the author's rebuttal, the reviewers still think that this is an incremental work, and the theory and experiments .are inconsistent. The authors are encouraged to consider the the reivewer's comments to improve the paper.